# Study of Dandelion (*Taraxacum mongolicum* Hand.-Mazz.) Salt Response and Caffeic Acid Metabolism under Saline Stress by Transcriptome Analysis

**DOI:** 10.3390/genes15020220

**Published:** 2024-02-09

**Authors:** Zhe Wu, Ran Meng, Wei Feng, Tassnapa Wongsnansilp, Zhaojia Li, Xuelin Lu, Xiuping Wang

**Affiliations:** 1Institute of Coastal Agriculture, Hebei Academy of Agriculture and Forestry Sciences, Tangshan 063299, China; wuzhe26@163.com (Z.W.); yoki_meng@163.com (R.M.); tofriendzhaojia@163.com (Z.L.); nkslxl@163.com (X.L.); 2Faculty of Science and Fisheries Technology, Rajamangala University of Technology Srivijaya, Trang 92150, Thailand; fengwei522106@163.com

**Keywords:** ethylene response, phenolic acid, salt-response genes, saline–alkali land, salt-tolerant plant

## Abstract

Utilizing salt-tolerant plants is a cost-effective strategy for agricultural production on salinized land. However, little is known about the mechanism of dandelion (*Taraxacum mongolicum* Hand.-Mazz.) in response to saline stress and caffeic acid biosynthesis. We investigated the morphological and physiological variations of two dandelions, namely, “BINPU2” (dandelion A) and “TANGHAI” (dandelion B) under gradient NaCl concentrations (0, 0.3%, 0.5%, 0.7%, and 0.9%), and analyzed potential mechanisms through a comparison analysis of transcriptomes in the two dandelions. Dandelion A had a high leaf weight; high ρ-coumaric acid, caffeic acid, ferulic acid, and caffeoyl shikimic acid contents; and high activities of POD and Pro. The maximum content of four kinds of phenolic acids mostly occurred in the 0.7% NaCl treatment. In this saline treatment, 2468 and 3238 differentially expressed genes (DEGs) in dandelion A and B were found, of which 1456 and 1369 DEGs in the two dandelions, respectively, showed up-regulation, indicating that more up-regulated DEGs in dandelion A may cause its high salt tolerance. Further, Gene Ontology (GO) analysis and Kyoto Encyclopedia of Genes and Genomes (KEGG) analysis showed that dandelion salt response and caffeic acid metabolism were mainly enriched in the phenylpropanoid biosynthesis pathway (ko00940) and response to ethylene (GO: 0009723). The caffeic acid biosynthesis pathway was reconstructed based on DEGs which were annotated to *PAL*, *C4H*, *4CL*, *HCT*, *C3′H*, and *CSE*. Most of these genes showed a down-regulated mode, except for parts of DEGs of *4CL* (TbA05G077650 and TbA07G073600), *HCT* (TbA03G009110, TbA03G009080, and novel.16880), and *COMT* (novel.13839). In addition, more up-regulated transcription factors (TFs) of ethylene TFs in dandelion A were found, but the TFs of *ERF104*, *CEJ1*, and *ERF3* in the two dandelions under saline stress showed an opposite expression pattern. These up-regulated genes could enhance dandelion salt tolerance, and down-regulated DEGs in the caffeic acid biosynthesis pathway, especially *CSE* (TbA08G014310) and *COMT* (TbA04G07330), could be important candidate genes in the synthesis of caffeic acid under saline stress. The above findings revealed the potential mechanisms of salt response and caffeic acid metabolism in dandelion under saline stress, and provide references for salt-tolerant plant breeding and cultivation on saline–alkali land in the future.

## 1. Introduction

China has a large area of saline–alkali land of approximately 100 million hectares. Soil salinization has seriously restricted agricultural production [1]. Culturing salt-tolerant plants is considered to be a cost-effective strategy for utilizing saline–alkali land [1,2] because moderate saline stress is beneficial for the accumulation of osmoprotectants and/or secondary metabolites in plants, thereby improving plant quality, especially in medicinal plants such as *Abelmoschus manihot* (L.) Medik and *Taraxacum mongolicum* Hand.-Mazz [1,2,3].

Osmoprotectants include amino acids, sugar, polyhydric alcohol, etc. These compounds accumulated in cells can improve plant stress tolerance, as they scavenge free radicals and activate protective enzymes [3]. Huang et al. reported that the application of proline (Pro) on cucumber was beneficial for the increase in leaf relative water content, Pro content, and peroxidase (POD) activity and the decrease in malondialdehyde (MDA) content, thereby decreasing the inhibitory effect of salt damage [4]. Phenolic compounds are important plant secondary metabolites. They play important roles in eliminating reactive oxygen species [5]. Phenolics have been found in various kinds of plants, especially in compositae.

Dandelion belongs to the *Asteraceae* family. It is rich in phenolic acids; has antibacterial, anti-inflammatory, and tumor-inhibiting effects; and is usually used in medicine and health care in China [1,6,7]. Dandelion can tolerate strong salt stress and is easily found in saline–alkali land [1]. According to the Chinese Pharmacopoeia (the 2015 edition), dandelion caffeic acid content shall exceed 0.02%. Current studies found that appropriate saline stress enhances the caffeic acid content in dandelion [1]. However, the mechanism of how saline stress enhances caffeic acid biosynthesis in dandelion is still insufficiently studied. 

Caffeic acid and its derivatives are synthesized in plants via the shikimic acid pathway from phenylalanine through deamination to form cinnamic acid, which is then derived from cinnamic acid [8]. However, their biosynthetic pathways in different plants or when under abiotic stress are different, mainly including two pathways: one is the synthesis of rosmarinic acid and salicylic acid using 4-coumaric acid and 4-hydroxyphenyllactic acid as precursors, which mainly exist in plants of the family *Labiatae*, represented by *Salvia divinorum*; and the other is the synthesis of chlorogenic acid compounds using quinic acid, caffeic acid, and ferulic acid as precursors, which are mainly found in the family *Lonicera*, represented by *Lonicera japonica* Thunb. During the biosynthetic pathway, enzymes such as phenylalanine ammonia lyase (PAL), quinate O-hydroxycinnamoyl transferase (HQT), caffeic acid O-methyltransferase (COMT), etc., regulate the synthesis of phenolic acids [6,9,10]. Meanwhile, researching the function of these enzyme-related genes has also become a focus. Following the publication of whole-genome sequences of dandelion, mining and researching caffeic acid and its derivative-related genes has become feasible, which could help to understand the mechanism of dandelion in response to saline stress [11,12].

Current studies mainly focus on the effects of abiotic factors on dandelion growth, biomass, chemical components, etc., while the mechanism of caffeic acid and its derivatives’ synthesis under saline stress is still not sufficiently understood. Hence, the potential mechanism of dandelion caffeic acid biosynthesis and salt response were investigated, aiming to provide references for dandelion breeding.

## 2. Materials and Methods

### 2.1. Materials and Design

Dandelion (*T. mongolicum* Hand.-Mazz.) resource A (*T. mongolicum* cv. “BINPU2”) and B (*T. mongolicum* cv. “TANGHAI”) were used in this research. Dandelion A was bred by our institute and dandelion B was a wild resource in the locality. Dandelion A performs better in leaf yield and phenolic acid content under saline stress than dandelion B. The dandelion breeding process and performance of the dandelion resources are described in [7].

The experiment was conducted in a rain-proof shelter at our institute of Hebei Academy of Agriculture and Forestry Sciences in Hebei Province, China in April of 2023. Dandelion seeds were sown in flower pots (height of 32 cm, and diameter of 28 cm) with a mixed substrate of perlite, vermiculite, and peat at a ratio of 2:1:1. Each pot had one healthy seedling with 5 leaves after germination of 25 days, and was then irrigated with 150 mL NaCl solutions every 2 days. NaCl concentrations were set at 0, 0.3%, 0.5%, 0.7%, and 0.9%, the first of which was the NaCl-free control treatment (CK). Each NaCl treatment was repeated 7 times. The temperature of the whole treatment period ranged from 25 ℃ to 32 ℃, and water relative humidity was maintained at around 65%. After 21 days of NaCl treatment, fresh and dry leaf weight (g) and leaf length (cm) were checked. The physiological parameters involving enzyme activity and phenolic acid content were also determined, as described in the following sections.

### 2.2. Phenolic Acid Examination

Caffeic acid (μg/g), ρ-coumaric acid (μg/g), ferulic acid (μg/g), and caffeoyl shikimic acid contents (μg/g) were determined based on the method described in [13,14]. Dandelion leaves were dried at 80 °C in an oven, and then were soaked in 80% methanol with a liquid-to-solid ratio of 15:1 and ultrasonic-extracted for 30 min. The obtained extracts were analyzed using ultra-high-performance liquid chromatography (Vanquish series UHPLC, Thermo Fisher Scientific Inc., Waltham, MA, USA). Details of the analytical conditions are described in the literature [15].

### 2.3. MDA, POD, and Pro Measurements

A total of 0.5 g fresh leaf sample was taken for the determination of the contents or activity of POD (U/g), MDA (μmol/g), and Pro (μg/g). The leaf samples were extracted and centrifuged following the instructions of the POD, MDA, and Pro assay kits (ml095259, ml093064, and ml094958; Shanghai Enzyme-linked Biotechnology Co., Ltd., Shanghai, China). Then, a microplate reader (Varioskan Lux, Thermo Fisher Scientific Inc., Waltham, MA, USA) was used to assay the three physiological parameters following the test kit instructions.

### 2.4. RNA-Seq Analysis

Fresh leaf samples were collected from a 0.7% NaCl treatment for RNA-Seq analysis because the highest contents of phenolic acids in the two dandelions were mostly found in this saline treatment based on the current experimental results. RNA-Seq was performed by Biomarker Technologies Co., Ltd. (Beijing, China). The sequence data were aligned to reference genome GWHBCHG00000000 (https://ngdc.cncb.ac.cn/gwh/Assembly/19733, accessed on 5 February 2024). Differentially expressed genes (DEGs) analysis, Gene Ontology (GO) analysis, and Kyoto Encyclopedia of Genes and Genomes (KEGG) analysis were carried out with default values using online tools at https://cloud.metware.cn. Details are referred to in the literature [16].

### 2.5. Real-Time Quantitative PCR (RT-qPCR) Validation

Eight DEGs were randomly selected to validate transcriptome data with the reference gene of GAPDH. Primers were designed by Primer 6.0 software (Appendix A). RT-qPCR was performed as described in the literature [17].

### 2.6. Data Processing

The experimental data were analyzed by one-way analysis of variance (ANOVA), and significant differences were determined by Duncan’s multiple range test using SPSS 22.0 Statistics (SPSS Inc., Chicago, IL, USA).

## 3. Results

### 3.1. Dandelion Morphological and Physiological Changes

The leaf weight and leaf length of dandelion A (BINPU2) and dandelion B (TANGHAI) in all the saline treatments significantly declined compared with the CK treatment after 21 days of NaCl solution treatment (Figure 1). Dandelion A had a higher leaf weight and leaf length than dandelion B in all saline treatments. The two parameters of dandelion A were significantly declined when the NaCl concentration was over 0.5%, while those of dandelion B were significantly declined in the saline treatment over 0.3%. However, the physiological indices involving Pro, MDA, and POD in the two dandelions were significantly changed with the saline treatment at 0.5% or 0.7%. Saline stress promotes the accumulation of protection substances such as POD and Pro, and harmful substances such as MDA in plants [18]. Dandelion A showed a higher content of POD and Pro and lower content of MDA than dandelion B. These results indicated that dandelion A has relatively strong salt tolerance.

### 3.2. Changes in Caffeic Acid and Its Derivatives

Dandelion A showed higher phenolic acid contents than dandelion B in all saline treatments (Figure 2). The contents of caffeic acid and ρ-coumaric acid increased following the increase in NaCl concentration in the two dandelions, and were highest with the 0.7% NaCl treatment in dandelion A and 0.9% NaCl treatment in dandelion B. Ferulic acid and caffeoyl shikimic acid overall showed similar variation trends with the saline treatment; however, their contents significantly varied with the saline treatment of 0.3%. 

### 3.3. Transcriptome Sequencing and Alignment

We analyzed the RNA-seq data from dandelion A and dandelion B under saline treatment (0.7% NaCl treatment), with three biological replicates. An average of 47.1 Mb of clean reads and 7.08 Gb of clean bases were obtained (Table 1). The clean read ratios were all above 97.42%. Q20 values all exceeded 97%. The *R*^2^ values within the same groups were all above 0.9 (Figure 3). The valid read rate of mapping between each sample and the reference genome exon region all exceeded 86.59% (Figure 3). 

### 3.4. Analysis of DEGs

The DEGs of dandelion A and dandelion B with CK vs. saline treatment were obtained (Figure 4): 2468 and 3238 DEGs for ACK vs. AS and BCK vs. BS, respectively, were identified, with 673 common DEGs, of which 1456 and 1369 were up-regulated genes in each group, implying that dandelion A could have higher salt tolerance. Moreover, we compared DEGs within the dandelion resources and found 9354 and 10148 DEGs in the groups of ACK vs. BCK and AS vs. BS, indicating a different salt tolerance or salt response in dandelion A and B.

### 3.5. GO Enrichment Analysis of DEGs

The results showed that the category of biological process mainly involved GO terms of cellular process, metabolic process, response to stimulus, etc.; the category of cellular component mainly involved the GO term of cellular process; and the category of molecular function mainly involved GO terms of binding, catalytic activity, transcription regulator activity, etc. Upon further analysis of the top 50 GO terms, the DEGs of the dandelion under saline stress were found to be highly enriched in the secondary metabolite biosynthetic process (GO:0044550), response to ethylene (GO:0009723), phenylpropanoid catabolic process (GO:0046271), cinnamic acid metabolic process (GO:0009803), apoplast (GO:0048046), hydrolase activity, hydrolyzing O-glycosyl compounds (GO:0004553), monooxygenase activity (GO:0004497), O-methyltransferase activity (GO:0008171), etc. (Figure 5 and Appendix A). The above enriched GO terms involved 298 and 305 DEGs in dandelion A and B under saline stress, of which 177 and 164 genes were up-regulated, respectively, in the two dandelions (Appendix A). These DEGs were closely related to the synthesis of phenolic compounds in the dandelion under saline stress.

### 3.6. KEGG Pathways of DEGs

All the DEGs for dandelion A and dandelion B under saline treatment (namely, ACK vs. AS, and BCK vs. BS) were assigned to 121 and 124 pathways in the KEGG database, respectively. Of these, the significantly enriched terms were plant hormone signal transduction (ko04075), MAPK signaling pathway (ko04016), metabolic pathways (ko01100), starch and sucrose metabolism (ko00500), biosynthesis of secondary metabolites (ko01110), and phenylpropanoid biosynthesis (ko00940) (Figure 6). These results indicated that biosynthesis of dandelion caffeic acid and its derivatives and salt response were mainly concentrated in the pathways of secondary metabolism and plant signal transduction under saline stress.

### 3.7. DEG Analysis of Phenylpropanoid Biosynthesis Pathway

Caffeic acid is a kind of secondary metabolite and the main bioactive component in dandelion. To reveal the salt response mechanism of caffeic acid biosynthesis, the DEGs of the phenylpropanoid biosynthesis pathway was studied through the KEGG pathway. We have summarized the DEGs related to caffeic acid biosynthesis in Appendix A. Dandelion A and dandelion B showed 20 and 21 DEGs in this pathway (Figure 7). These DEGs were expressed in the path from cinnamic acid to caffeoyl shikimic acid, which were annotated to *PAL*, *C4H*, *4CL*, *HCT*, *C3′H*, and *CSE*. Most of these DEGs exhibited down-regulation under saline stress, except for parts of DEGs in the two dandelions that exhibited opposite expression patterns, including the following:

Six DEGs of *PAL* in the two dandelions all showed a down-regulated pattern, but two DEGs of *4CL* in two dandelions simultaneously expressed an up-/down-regulated pattern during the path of cinnamic acid to cinnamoyl-CoA. Similarly, different quantities of *HCT* genes in the two dandelions also showed an opposite regulation pattern simultaneously. Most notably, *CSE* was significantly down-regulated under saline treatment in dandelion A, but non-significantly expressed in dandelion B. In addition, two *COMT* genes in dandelion A expressed an opposite regulation pattern, but four *COMT* genes in dandelion B all expressed a down-regulated pattern. Therefore, we speculate that the different DEGs in dandelion A and B, especially *COMT* and *CSE*, played important roles in the synthesis of caffeic acid compounds and the response to saline stress.

### 3.8. Transcription Factors (TFs) and Ethylene Response

TFs in the two dandelions under saline stress were expressed. Most TFs were classified into the families of MYB, C2H2, WRKY, bHLH, ERF, LOB, and bZIP. The quantity of TFs in the ERF, C2H2, and MYB families were all more highly expressed in the two dandelions (Figure 8). A total of 181 and 162 TFs in dandelion A and B were specifically expressed under saline stress. Among them, there were 109 and 62 up-regulated genes in the two dandelions, respectively (Figure 8). 

TFs typically work together with signaling pathways to actively cope with saline stress [19,20]. We identified all the DEGs related to ethylene response (Figure 9). The ethylene transcription factors *ERF104* and *CEJ1* in dandelion A and *ERF3* and *ERF8* in dandelion B were significantly up-regulated under saline stress, but *ERF104*, *ERF3*, and *CEJ1* showed an opposite expression mode in the two dandelions (Figure 9), indicating that the three TFs may use bidirectional regulation for saline stress or be activated by multiple abiotic factors such as ABA and jasmonic acid [19,20].

### 3.9. Verification of RNA-seq Data

We performed RT-qPCR analysis with the same RNA-seq samples to validate the transcriptome data. A total of eight DEGs enriched in the phenylpropanoid biosynthesis pathway were randomly selected (Appendix A). The results of RT-qPCR for the eight DEGs showed the same expression pattern as the RNA-seq data (Figure 10). 

## 4. Discussion

Soil salinization limits crop production, ultimately making food security a priority consideration for agricultural development in many countries [21]. Utilizing salt-tolerant plants is a very cost-effective strategy for the utilization of saline–alkali land [22]. Dandelion is a prosperous and widely used medicinal plant in China. It can be grown in saline–alkali land. Here, we discuss the differences in the salt response and caffeic biosynthesis between the two dandelions to explore the mechanism of dandelion salt tolerance and caffeic acid biosynthesis under saline stress.

### 4.1. Morphological and Physiological Adaptation to Saline Stress

Plants undergo a series of complex morphological and physiological changes under saline stress [6]. Saline stress significantly restricts plant growth at the bud and seedling stages, and finally affects the yield and quality [23]. But some halophytes such as *Mesembryanthemum crystallinum* L. have higher yield and better growth under a moderate soil salt content [24]. In this study, dandelion leaf weight and phenolic acid compounds showed an increasing pattern with increasing salt treatment. Dandelion A exhibited higher salt tolerance than dandelion B (Figure 1). This could be attributed to enhanced physiological regulation of saline stress due to the high level of Pro and POD in dandelion A under saline stress. 

A water deficit and excessive Na^+^ and Cl^−^ content in plant cells are the main causes of salt damage. The excessive irons accumulated in the cells cause osmotic and ionic stress in plants [25]. Osmotic and ionic stress also lead to ROS production [26]. To cope with the oxidative damage, plants can activate the protection enzymes to eliminate ROS. These protection enzymes include superoxide dismutase (SOD), peroxidase (POD), ascorbate peroxidase (APX), catalase (CAT), etc. [27]. Hence, the levels of these parameters including morphological indices could reflect a certain plant’s salt tolerance [1,2]. A report shows that dandelion could tolerate 1.5% NaCl and its salt tolerance threshold was around 0.43% based on morphological parameters [1]. In this study, the morphological and physiological parameters in dandelion A significantly changed after the treatment from 0.5% to 0.7% NaCl, and dandelion B after 0.3% to 0.5% NaCl (Figure 1), which meant that dandelion A had a higher salt tolerance threshold.

Saline stress promotes plants to convert carbohydrates into secondary metabolites [6]. Reports show that some plants such as *Carthamus tinctorius* L can adapt to saline stress by accumulating phenolic compounds [25,27]. We checked the contents of caffeic acid, ρ-coumaric acid, ferulic acid, and caffeoyl shikimic acid and found that they increased following the saline treatment, and the maximum phenolic acids content was found with the NaCl treatment of 0.7–0.9% (Figure 2). This result was consistent with the report that moderate salt stress was beneficial for the production of phenolic acid in dandelion [1]. 

### 4.2. DEGs in Dandelion under Saline Stress 

The genus *Taraxacum* contains more than 3000 species [11]. Dandelion commonly exhibits apomixes [28]. Thus, the variation in genomes among dandelions is very complex [12]. We used RNA-Seq technology with the published dandelion genome as a reference to analyze the DEGs of the two dandelions under saline stress (Table 1). This method could identify salt-response genes as accurately as possible. We found a total of 2468 and 3238 DEGs in dandelion A and dandelion B, respectively, under saline stress (Figure 4). We confirmed that caffeic acid biosynthesis in dandelion was mainly involved in the phenylpropanoid pathway via GO function and KEGG function enrichment analysis [6,9,10], and reconstructed the biosynthesis path of caffeic acid under saline stress (Figure 7). 

Phenylalanine ammonia lyase (PAL) is the first key enzyme involved in the phenylpropanoid metabolism pathway for the biosynthesis of phenolic compounds in plants [5,29]. High saline treatment could induce an increase in PAL activity [5]. However, we found that all the DEGs of *PAL* in the two dandelions showed a down-regulated mode under saline stress (Figure 7). There were similar findings that 10 *PAL* genes in *Lotus japonicus* all exhibited down-expression under long-term NaCl treatment [30]. Thus, we suppose that long-term salt stress has led to the accumulation of massive synthetic substrates such as phenylalanine, and that the plant, in order to regulate the synthesis of downstream products, inhibits the expression of *PAL* genes to reduce enzyme activity, thereby playing a rate-limiting role. Similarly, most DEGs of *C4H*, *4CL*, *HCT*, *C3′H*, and *CSE* exhibited a down-regulated mode (Figure 7 and Appendix A), and they also played a similar rate-limiting role. 

Interestingly, some DEGs, especially more HCT genes in dandelion B, exhibited an up-regulated mode under saline stress. These up-regulated genes involved *4CL*, *HCT*, and *COMT*, but all exhibited far lower expression levels than the down-regulated genes except for *COMT* (Figure 7 and Appendix A). 4-Coumarate: CoA ligase (4CL) regulates the conversion of coumarin to different branching products in the phenylpropanoid metabolism pathway. These downstream products involve caffeic acid, cinnamic acid, and o-coumaric acids, etc., but most *4CL* gene functions are still unknown [31]. Liu et al. isolated three *4CL* genes from *Peucedanum praeruptorum* and found that the *4CL* genes significantly exhibited an up-regulated pattern with the treatment of MeJA, but were less expressed or down-regulated under NaCl treatment [31].

Hydroxycinnamoyl transferase (HCT) is generally related to lignin biosynthesis. Overexpression or inhibition of *HCT* genes could alter lignin composition [32]. Some plants also accumulate more lignin in cell walls to cope with salt damage [33]. Chowdhury et al. found that *HcHCT* from kenaf plant was positively expressed under long-term NaCl treatment. Also, they found that lignification clearly occurred in *Brassica oleracea* root under saline treatment [32,34]. Given the above discussion, we infer that up-regulated *4CL* and *HCT* have more effect on downstream product lignin synthesis than caffeic acid under saline stress.

Moreover, there are three notable genes in the caffeic acid biosynthesis path, namely, *C3H*, *CSE*, and *COMT*. ρ-Coumarate 3-hydroxylase (C3H) can convert p-coumaric acid into caffeic acid. However, we did not find it significantly expressed in the two dandelions under saline stress, but *CSE* was significantly down-expressed in dandelion A (Figure 7). A study reported that recombinant *CSE* could convert caffeoyl shikimate into caffeic acid efficiently in vitro [35]. Compared with upstream genes such as *PAL, HCT, 4CL,* etc., *CSE* may be the key rate-limiting enzyme directly influencing the synthesis of caffeic acid. Due to this, dandelion A exhibited higher salt tolerance and caffeic acid content than dandelion B. And combined with the above discussions of upstream genes, it could be inferred that dandelion A could clearly accumulate massive synthetic substrates under long-term salt stress; in order to regulate the synthesis of downstream metabolites or maintain a certain physiological balance, more upstream enzyme-related genes were inhibited to limit enzymes to synthesize excessive downstream metabolites. However, this hypothesis needs to be further verified.

Contrary to CSE, caffeic acid o-methyltransferase (COMT) is a key rate-limiting enzyme converting caffeic acid to ferulic acid. Up-regulation of *COMT* enhances the salt stress tolerance of plants. Sun et al. found that overexpression of *SlCOMT1* could maintain the balance of Na^+^/K^+^ in tomato under salt treatment and up-regulate some stress-related genes, such as *WRKY33*, *MAPK*1, *AREB1*, etc., thereby enhancing salt tolerance. [36]. We found two *COMT* genes (novel.13839 and TbA04G072330) in dandelion A exhibiting up-regulation and down-regulation patterns, respectively, while all four *COMT* genes in dandelion B showed down-regulation patterns (Figure 7). Thus, we speculate that the up-regulated *COMT* enhanced dandelion salt tolerance, and down-regulated *COMT* was favored for the accumulation of caffeic acid. However, the functions of the specific DEGs of *COMT* that are related to salt tolerance or caffeic acid synthesis still need to be verified.

### 4.3. Specific Expression of TFs

GO and KEGG pathway enrichment analyses showed that the DEGs of dandelion under saline stress were annotated to response to ethylene (GO: 0009723) and plant hormone signal transduction (ko04075) (Appendix A). The synthesis of ethylene in plants is closely related to abiotic stress. The TFs related to ethylene transcription factors in the two dandelions were mainly enriched in the ERF family (Figure 8). ERF transcription factors regulate the expression of ethylene-responsive genes, thereby participating in plant development and stress response [37].

*ERF104* is a substrate of MAPK 6 and affects some genes encoding TFs such as *WRKY25*, *ZAT6*, *ERF-1*, etc. [20]. *ERF104* can regulate the expression of salt-tolerant genes in different signaling pathways, thereby enhancing plant salt tolerance [38]. Overexpression of *ERF3* also significantly improved salt tolerance in wheat [39]. ERF8 induces ethylene biosynthesis. It was up-regulated under salt stress in *Arabidopsis thaliana* [20]. *CEJ1* transcript behaves like *ERF8*, and both respond to ethylene and ABA. But *CEJ1* responds more to ABA [19]. In this study, *ERF104* and *CEJ1* in dandelion A showed significant up-expression under saline stress, but down-expression in dandelion B, and *ERF3* and *ERF8* showed up-expression in dandelion B, but *ERF3* showed down-expression in dandelion A (Figure 9), indicating that dandelion A and dandelion B had different salt-response mechanisms. Given the above discussion, the higher number of TFs up-regulated in dandelion A may be one of the reasons for its high salt tolerance.

## 5. Conclusions

The potential mechanisms of salt response and caffeic acid metabolism in dandelion under saline stress were investigated through a comparison analysis of transcriptomes with the reference genome of the two dandelions. Morphological and physiological parameters analysis under gradient NaCl solution treatments confirmed that dandelion A (BINPU2) had higher salt tolerance and phenolic acid accumulation than dandelion B (TANGHAI). Transcriptome analysis found different quantities of DEGs in the two dandelions and more up-regulated DEGs in dandelion A. Further GO and KEGG analyses of the DEGs found that dandelion caffeic acid metabolism and salt response were mainly enriched in the phenylpropanoid biosynthesis pathway (ko00940) and response to ethylene (GO: 0009723). We reconstructed the caffeic acid biosynthesis pathway based on the DEGs, showing that most rate-limiting enzyme-related genes exhibited a down-regulated pattern. Among these down-regulated genes, *CSE* (TbA08G014310) and *COMT* (TbA04G07330) could be the key genes regulating the metabolism of caffeic acid under saline stress. For the response to ethylene, more up-regulated TFs of ethylene transcription factors were found in dandelion A; however, the TFs of *ERF104*, *CEJ1*, and *ERF3* in the two dandelions showed opposite expression patterns. Given the above DEG analysis, it can be inferred from the gene responses that the two dandelions have different salt-response mechanisms, and that more up-regulated genes in dandelion A may cause its higher salt tolerance.

## Figures and Tables

**Figure 1 genes-15-00220-f001:**
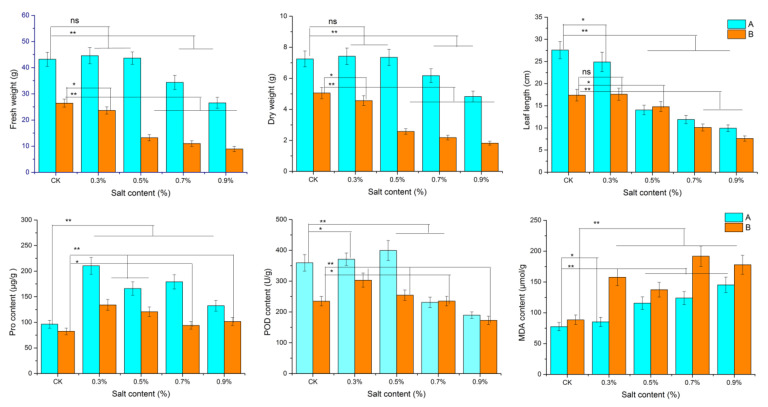
Performance of two dandelions under different NaCl treatments. Blue represents dandelion A (BINPU2), and orange represents dandelion B (TANGHAI) after 21 days of NaCl treatment. Error bars are standard deviations, *n* = 7. One-way ANOVA with Duncan’s multiple range test was used to compute *p*-values; ** *p* < 0.01, * *p* < 0.05; ns, no significant difference.

**Figure 2 genes-15-00220-f002:**
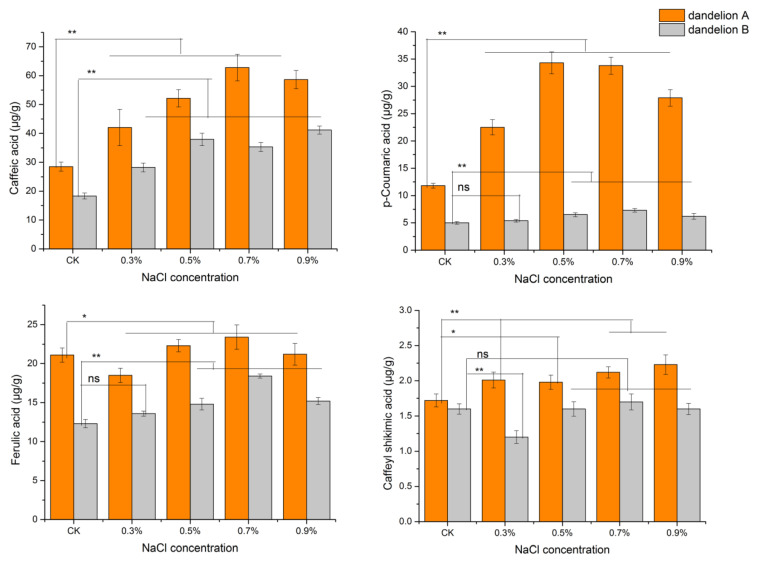
Phenolic acid contents of two dandelions after 21 days of NaCl treatment. Error bars represent standard deviation, *n* = 7. One-way ANOVA with Duncan’s multiple range test was used to compute *p*-values; ** *p* < 0.01, * *p* < 0.05; ns, no significant difference.

**Figure 3 genes-15-00220-f003:**
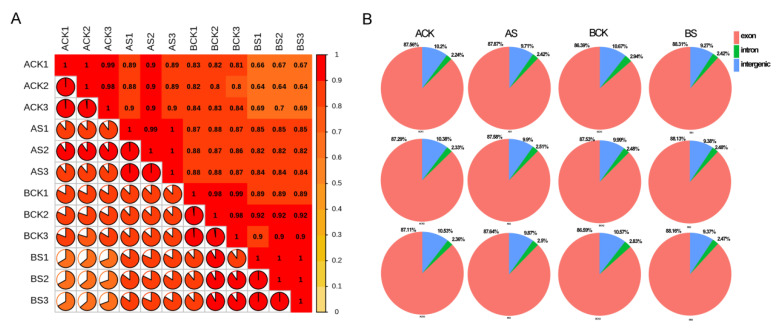
Sequencing quality analysis for two dandelions. (**A**) Pearson’s correlation coefficient (*R^2^*) between different samples. (**B**) Reads comparison region distribution.

**Figure 4 genes-15-00220-f004:**
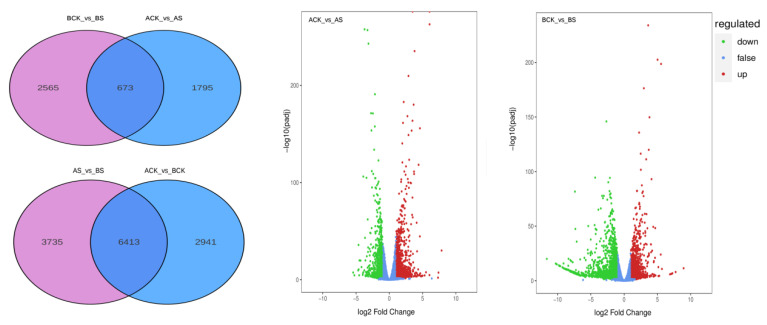
Venn and volcano plot of DEGs within groups of dandelion A (BINPU2) and dandelion B (TANGHAI) with CK vs. saline treatment (S).

**Figure 5 genes-15-00220-f005:**
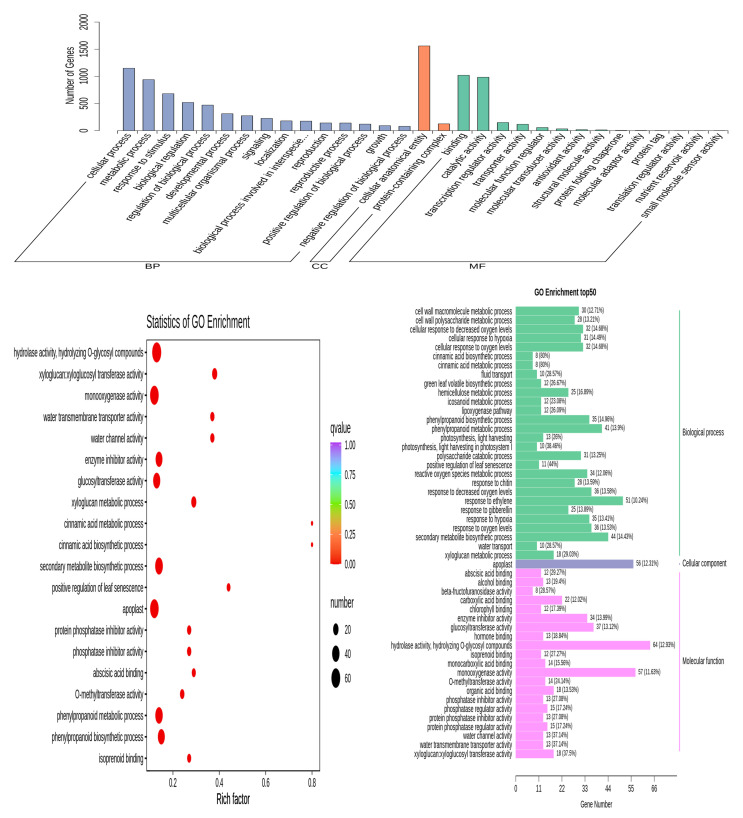
Go term classification and enrichment analysis of DEGs for dandelion group of ACK vs. AS.

**Figure 6 genes-15-00220-f006:**
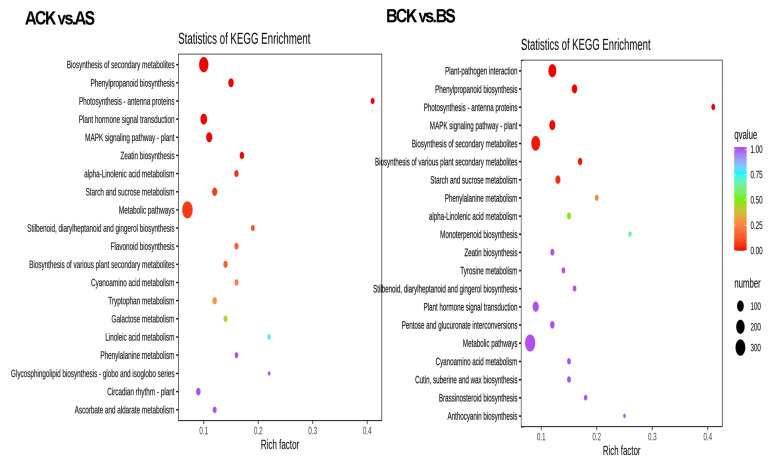
KEGG enrichment analysis of DEGs for two dandelions.

**Figure 7 genes-15-00220-f007:**
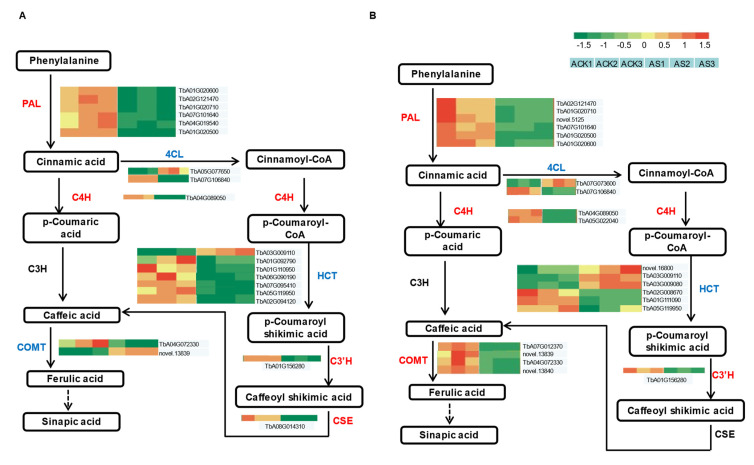
Caffeic acid biosynthesis pathway in two dandelions under saline stress. (**A**) Dandelion A (BINPU2), (**B**) dandelion B (TANGHAI).

**Figure 8 genes-15-00220-f008:**
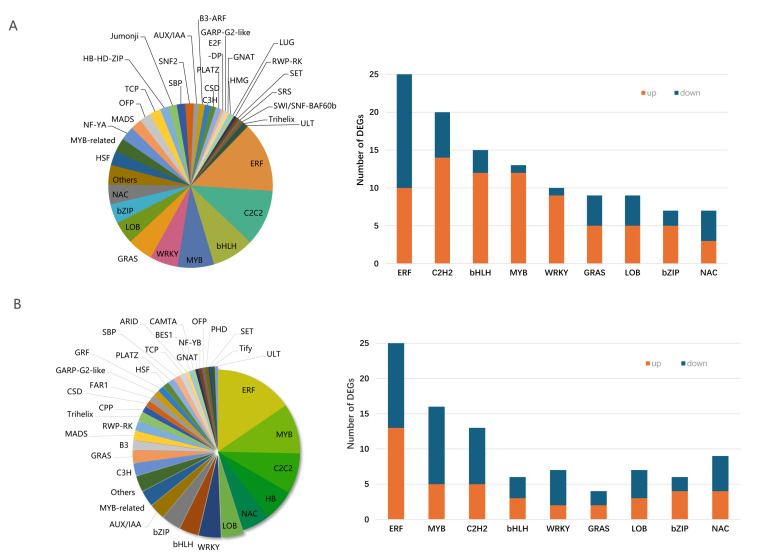
Expressed TFs in two dandelions under saline stress. (**A**) and (**B**) represent dandelion A (BINPU2) and dandelion B (TANGHAI); the left column is the quantity of enriched TF families for the two dandelions; and the right column is the number of DEGs in the main TF families for the two dandelions.

**Figure 9 genes-15-00220-f009:**
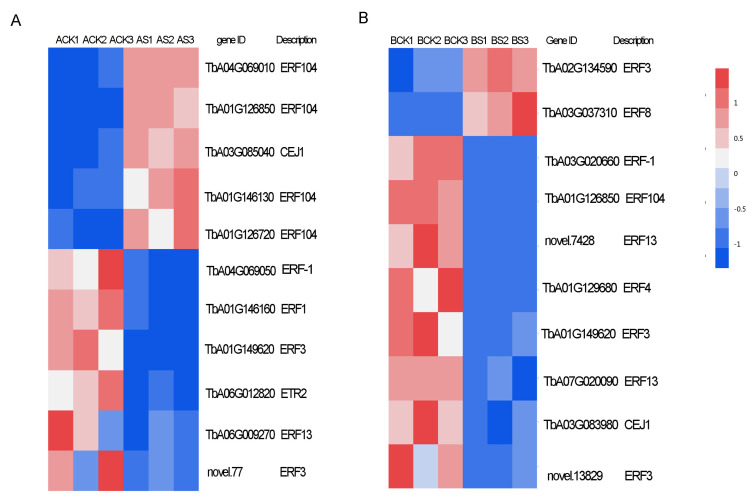
Heatmap of TFs related to ethylene response for two dandelions. (**A**) and (**B**) represent dandelion A (BINPU2) and dandelion B (TANGHAI).

**Figure 10 genes-15-00220-f010:**
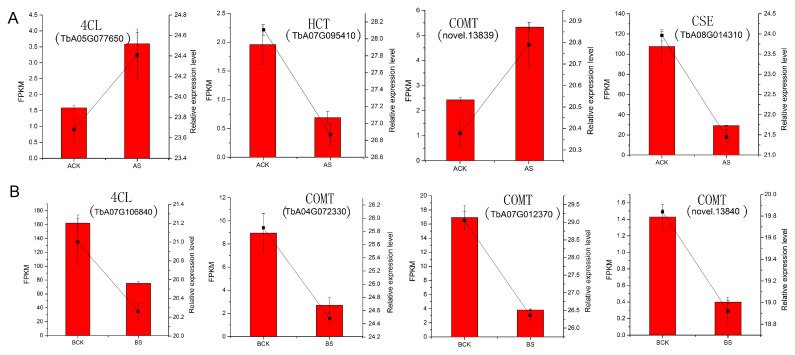
Expression patterns of eight genes in two dandelions. (**A**) Dandelion A (BINPU2), (**B**) dandelion B (TANGHAI); the bar graphs represent the RNA-seq data (FPKM), and the line graphs represent the RT-qPCR result. Data are mean ± SD, *n* = 3.

**Table 1 genes-15-00220-t001:** Summary of RNA-seq data of 12 libraries.

	Raw Reads (Mb)	Clean Reads (Mb)	Clean Base (G)	Q20 (%)	GC Content (%)	Mapped Reads	Mapped Ratio
ACK ^1^	47.35 ± 1.21	46.27 ± 1.02	6.94 ± 0.15	97.41 ± 0.08	43.69 ± 0.15	35.95 ± 0.73	0.78 ± 0.003
AS ^1^	47.39 ± 0.48	46.29 ± 0.3	6.94 ± 0.05	97.33 ± 0.12	43.49 ± 0.16	36.27 ± 0.18	0.78 ± 0.001
BCK ^1^	50.39 ± 1.5	48.86 ± 1.83	7.33 ± 0.27	97.39 ± 0.04	43.81 ± 0.54	42.13 ± 1.73	0.86 ± 0.003
BS ^1^	48.65 ± 0.83	47.35 ± 0.83	7.10 ± 0.12	97.34 ± 0.15	44.52 ± 0.16	40.60 ± 0.7	0.86 ± 0.002

^1^ ACK, AS, BCK, and BS are the treatments of dandelion A (BINPU2) or dandelion B (TANGHAI) under non-saline stress (CK) and saline stress (S), and all of the group symbols in this text have these same definitions. Data in each group are shown as mean ± standard deviation, *n* = 3.

## Data Availability

Data are contained within the article and Appendix A.

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
