# Peer review of "Study of Dandelion (Taraxacum mongolicum Hand.-Mazz.) Salt Response and Caffeic Acid Metabolism under Saline Stress by Transcriptome Analysis"

_genes, 2024, doi:10.3390/genes15020220_

Round 1

Reviewer 1 Report

Comments and Suggestions for Authors

I made some suggestions, please the attached file. 

Comments on the Quality of English Language

Author Response

     On behalf of all the authors, I greatly appreciate your meticulous review work, which has greatly helped to improve the quality of this paper. We have revised the contents according to your valuable suggestions and responded to all the questions you raised. Thank you again.

  1. recommend to improve the general structure of introduction, in order to be more readable and concise.

Reply: we have modified this section and deleted some redundant narration, see the revised version.

  1. I suggest to specify in which kind of plants.

Reply: we have added the details; see the revised version (Introduction, paragraph 1).

  1. Check citing style

Reply: we have revised all citing style for this journal.

  1. Because you compare Dandelion A vs B, I recommend to give more details about it. Are really the same plant variety? How do you determine the cultivar? Is there a research article about it? Cite it. (Materials and Design section)

Reply: we have added some necessary details about the two dandelions and give the citations.

Dandelion A is a certified variety which was bred by our institute; dandelion B is a wild resource collected in the locality. Dandelion B is used as comparison, as its lower phenolic acids and leaf yield under saline stress. So the two dandelions are not the same plant variety. Actually, before this study, we have compared more than 86 dandelion resources (some are collected, some are created by NaCl-induced method). Then we obtained a high-phenolic-acid-producing dandelion line by repeatedly screening and identification through various methods. All the related information can be found from our publications (here give some links: (a) https://kns.cnki.net/kcms/detail/46.1068.S.20230309.1716.008.html (b) https://www.cnki.com.cn/Article/CJFDTotal-JSNY202119034.htm  (c) https://www.mdpi.com/2311-7524/8/12/1167 ).

  1. Indicate the volume or quantity of each one. How much days after germination? Verify units (NaCl concentration). DEGs is not a determined parameter included at this sentence. Why not? fresh and dry weight or yield? Why only the 0.7% treatments were sampled? (Material and Method----Materials and Design section)

Reply: we have supplemented the necessary information according to your valuable suggestions. Details were seen in the revised version. Here we select the 0.7% treatment as the only sampled for RNA-Seq analysis, mainly because the maximum phenolic acid content in two dandelions occurred in this treatment. We have supplemented this description including the description of DEGs in the section of “RNA-Seq analysis”.

  1. Please be more specific. You need to mention the expressed units to measure them. (Material and Method---- MDA, POD and Pro Measurements section)

Reply: we have corrected this section.

  1. May be it could be added an image with A and B morphology results. Growth conditions were similar. Growth results of the evaluated parameters not. Please rewrite this sentence. Be more clear. (Results----3.1 section)

Reply: we think the result description is enough for supporting the further transcriptome analysis. For this study, we mainly focus on the transcriptome analysis to reveal the mechanism of salt-response and caffeic acid biosynthesis under saline stress. We have revised the results analysis and rewritten the related sentences. See the revised version.

  1. Figure or Table captions: dandelion A (BINPU2) and dandelion B (TANGHAI). at n days after germination.

Reply: we have checked all figure captions, and corrected them all.

  1. be more specific ... Positive response or negative response of Dandelion evaluated parameters after o.5% or what? May be it could be worthy to mention it at the beginning of MM section after lines 179-181. (Results----3.2 section)

Reply: we have added details, see the revised version.

We recognized your valuable suggestions; we have moved this part to Material and Method section. See the section of “RNA-Seq analysis”.

  1. 20 DEGs are similar? which ones are different... CSE was only the different? (Results----3.7 section)

Reply: our expression is confusing, we would like to say the quantities of DEGs in two dandelions were different, we have rewritten this section, see the revised version.

  1. Be aware to mix results and discussion (Results----3.8 section)

Reply: we have separated the result and discussion.

  1. What level of stress? low, high or moderate saline stress ... (Discussion)

Reply: we have given the specified description and concise this section.

  1. My comment on introduction at lines 42 to 45 is about this (see question 2). Results? Add references. (Discussion---4.1 section)

Reply: the same reply as question 12. We simplified this section.

  1. Be aware to no repeat results. Good discussion, Could you give more details? (Discussion---4.2 section)

Reply: refer to the question 11, we have separated the result and discussion. In addition, we give necessary details of discussion in this section. See the revised version.

  1. You need to improve your conclusions to be more concise.

Reply: we have refined this section.

  1. Others (References)

Reply: we have corrected citation style and spell errors.

Reviewer 2 Report

Comments and Suggestions for Authors

The text refers to the comparison of two dandelion plants and their response to exposure to different concentrations of NaCl through their transcriptome and the production of secondary metabolites (anthocyanins).

The work is of interest, however there are some details that could improve it: a. Arrange the keywords in alphabetical order and without repeating those in the title. b. Some of the images in the Figures are not clear, perhaps it would be appropriate to prioritize those images with the gene registration for example, write the names of the genes (i.e. Figures 9 and 10), enlarge figure 5. c. Some images and figures may be included in the annexes to allow a better explanation of the relevant data. d. A general diagram of the effect of the saline condition on the plant will allow a better understanding of the effect.

Author Response

On behalf of all the authors, I greatly appreciate your meticulous review work, which has greatly helped to improve the quality of this paper. We have revised the contents according to your valuable suggestions and responded to all the questions you raised. As follows:

  1. Arrange the keywords in alphabetical order and without repeating those in the title.

Reply: we have selected suitable keywords.

  1. Some of the images in the Figures are not clear, perhaps it would be appropriate to prioritize those images with the gene registration for example, write the names of the genes (i.e. Figures 9 and 10), enlarge figure 5.

Reply: we have modified the figures. Also provided the original images.

  1. Some images and figures may be included in the annexes to allow a better explanation of the relevant data.

Reply: thanks for your suggestions, we have edited some images in the annexes.

  1. A general diagram of the effect of the saline condition on the plant will allow a better understanding of the effect.

Reply: we have refined the discussion on the effect of the saline condition on the plant, and kept necessary descriptions. See the revised version.

Thank you again for your valuable suggestions!

Reviewer 3 Report

Comments and Suggestions for Authors

Very intersting research about the salt-response and caffeic acid metabolism under saline stress 3 by transcriptome analysis. The manuscript is well-written. I have only few comments in the attached document.

Author Response

On behalf of all the authors, I greatly appreciate your meticulous review work, which has greatly helped to improve the quality of this paper. We have revised the contents according to your valuable suggestions and responded to all the questions you raised. Thank you again.

  1. It is a very long sentence and confusing for the reader. Split it. (Abstract and Introduction section). Also, when for the first time use "dandelion' add Taraxacum mongolicum".

Reply: we have re-write these parts, see the revised version.

  1. Before the results, describe briefly your experiment. You referred to 0.7% NaCl treatment but the reader do not know the other treatments. (Abstract section)

Reply: we have added the necessary experimental details.

  1. You use for the first time the abbreviations, add the meaning

Reply: we have added the full name for all abbreviations.

  1. Rewrite this sentence, it is confusing (Abstract section)

Reply: we have re-write this sentence, see the revised version.

  1. It is better to use the scientific name

Reply: we have checked all scientific name of plants.

  1. What is this (CK) ? Why only this (0.7% NaCl treatment)?

Reply: here CK means solution free of NaCl, used as control. We have supplemented the details.  Here we select the 0.7% treatment as the only sampled for RNA-Seq analysis, mainly because the maximum phenolic acid content in two dandelions occurred in this treatment. We have supplemented this description in the section of “RNA-Seq analysis”.

  1. I think that it is better to refer this in the Material and Method section because it is not clear why you select this treatment for further analysis. (Results section)

Reply: as replied in last question, we recognized your valuable suggestions, and we have moved this description to Material and Method section.

Thank you for your suggestions.

Round 2

Reviewer 1 Report

Comments and Suggestions for Authors

The authors have attended to all observations I made. I reccommend to accept it in current form.